# Membrane Separation in the Nickel-Contaminated Wastewater Treatment

Magdalena Lech *, Oliwia Gala, Karolina Helińska, Katarzyna Kołodzińska, Hanna Konczak, Łukasz Mroczyński and Eliza Siarka

Department of Micro, Nano and Bioprocess Engineering, Faculty of Chemistry, Wroclaw University of Science and Technology, Wybrzeze Wyspianskiego 27, 50-370 Wroclaw, Poland
* Correspondence: magdalena.lech@pwr.edu.pl; Tel.: +48-71-3202803

**Abstract:** Nowadays, electroplating plants are factories that use huge amounts of water in the coating process of anti-corrosion layers. They are required to decrease the heavy metal content to very low values before releasing the post-process water into the aquatic environment. They very often decrease their content using coagulation combined with flocculation. However, these processes are often not effective enough, and the concentration of a given metal does not reach a satisfactory low level. The use of membrane techniques to purify this type of wastewater leads to a reduction in the content of heavy metals, including nickel, to zero values. This allows for not only reducing the negative impact on the aquatic environment but also a step toward more conscious management of water resources—namely, the reuse of water in the electroplating process. The following review not only describes the membrane methods used to treat the wastewater considered, e.g., nanofiltration, ultrafiltration, or electrodialysis, but also shows the directions of development of these processes.

**Keywords:** nickel; wastewater; complexing ultrafiltration; electrodialysis; selective electrodialysis

## 1. Introduction

The process of coating steel surfaces with nickel found its application in electroplating at the turn of the 19th and 20th centuries. Nowadays, according to the Nickel Institute [1], the nickel-plating process uses about 150,000 tons of nickel per year, and the greatest development of this technology has been in the last 50 years. The nickel-plating process is an important part of the metal processing industry, but it also causes serious environmental problems due to the massive discharge of effluents containing nickel and other heavy metals [2]. During the nickel-plating process, two main types of effluents are generated: the effluents from the electrolytic bath and the rinsing water. The composition and quantity of the resulting wastewater are influenced by many factors, such as the type of bath used, the quantity of rinse water, and the system of plant operation. The concentration of nickel ions in rinse water is in the range of 4–490 mg/L [3–5]. Nowadays, the trend in the utilization process development of this wastewater is regeneration and then returning to the process. The result of such actions is obtaining clean water, which is then returned to the baths with rinsing water and a concentration of metal ions, which can fill the losses with an electrolytic bath. This is particularly important not only because of the reduction of the negative impact on the environment and human health but also due to the amount and value of metals lost in the process. According to the Regulation of the Minister of Environment enclosed in the Journal of Laws of the Republic of Poland [Dz. U. z 2014 r., poz.1800], the maximum allowable nickel content in industrial wastewater that is removed from the factory must have a concentration lower than 0.5 mg/L.

The reduction of high concentrations of nickel ions in wastewater entering the treatment plant is also important for the proper growth and functioning of activated sludge, which is responsible for the decomposition of organic matter. On the other hand, numerous studies confirm that wastewater with a low $Ni^{2+}$ content (less than 5 mg/L) causes an

increase in activated sludge and improves its physicochemical properties. Gikas [6], in his work, showed that the amount of activated sludge increases with nickel concentrations, reaching about 27 mg/L. Above this value, a decrease in activity and proliferation of microorganisms is observed, while after exceeding 160 mg/L $Ni^{2+}$, a complete lack of their growth is noted.

Based on the above, from an economic and environmental point of view, regeneration, purification, and recycling of water contaminated by metals (including nickel) are always justified. Methods for appropriate wastewater treatment should be inexpensive, clean, rapid, and environmentally safe. These methods include chemical precipitation, coagulation, flocculation, filtration, ion-exchange, membrane separation processes, e.g., reverse osmosis, or electrodialysis [7]. For cases when the concentration of $Ni^{2+}$ ions is higher than 100 mg/L, coagulation and flocculation are both quite effective and cost-efficient methods [8]. However, the final concentration usually does not reach the demanded value (lower than 1 mg/L).

The mentioned membrane techniques, which are very popular in wastewater treatment processes of various origins, are potentially optimal solutions. Mainly because of the lowest environmental impact (no generation of chemical waste), high yields, and obtaining a high-quality product. This research presents a brief overview of membrane techniques used in the treatment of wastewater contaminated with metals, including nickel.

Membrane technics are the most promising methods for wastewater treatment and have been developed widely nowadays. The most popular are voltage-based techniques since nickel exists as a cation that can be transported through ion-exchange membranes. On the other hand, it is a very small structure that, in the case of pressure techniques, can be retained by RO membranes, at least by NF. These are completely different separation techniques, even though they use membrane structures. Putting all these different membrane-based techniques together in one place will allow you to compare them, choose the right techniques, and see their advantages and disadvantages.

The separation of heavy metals is also the reason for the development of complex ultrafiltration as well as the modification of other already known methods, so the presentation of these newer solutions in this article will allow us to notice the directions of development of membrane techniques in this area. This aims at increasing metal retention and the stream of treated wastewater.

Many papers considered different divalent ions. Most of them are trace metals such as Cu, Co, and Zn. Considerably fewer papers deal with the separation of nickel; however, considering the fact that the presented methods are successful for other divalent transition metals, it is highly probable that they will also be suitable for the separation of nickel ions from plating processes.

## 2. Membrane Technics Using in the Nickel Removing Process

Membrane separation is the process of selective transport of different compounds and chemicals through a porous or nonporous membrane structure, which separates two phases. The driving force of the process is the pressure, concentration, or temperature difference on both sides of the membrane. The pressure-driven membrane separation results in two streams: a retentate containing the retained molecules, which is returned to the process, and a permeate consisting of the purified water [9].

### 2.1. Nanofiltration

Nanofiltration (NF) or reverse osmosis (RO) are typically used to produce water suitable for reuse. In addition to removing impurities, reverse osmosis strips the water of all minerals. However, nanofiltration is more interesting for industrial researchers and technologists because it is more cost-effective than RO. It is characterized by good efficiency, lower than RO transmembrane pressure, and therefore low energy consumption. The working pressure of reverse osmosis is 1.5–10.5 MPa, while the working pressure of the nanofiltration membrane is 0.1–0.2 MPa [10]. A nanofiltration membrane is expected to

have pore diameters in the range of 1–2 nm and a molecular weight cutoff (MWCO) for neutral solutes in the range of 150 Da to 2000 Da [11]. Recently, many membranes have been made by modifying ultrafiltration membranes with polymers. There are modification methods such as interfacial polymerization (IP), physical mixing, layer-by-layer coating (LbL), and surface grafting.

Brooms Thabo et al. [12] compared three commercially available nanofiltration membranes. They chose two polyamide membranes: NF90 (MWCO = 150 Da), NF270 (MWCO = 200 Da), and one polypiperazine, PP-NF (MWCO = 150 Da). Separation of heavy metal salts at two concentrations showed that the NF90 membrane worked best, reached separating efficiency—98% of the nickel from the feed, while the NF270 membrane retained 70% and the PP-NF (polypiprrazine) 75%. This showed that MWCO is as relevant as membrane material. 150 Da NF membranes rejected the nickel to different extents due to this fact. The NF membranes cannot be sufficient sometimes to reject small compounds such as $Ni^{2+}$. The researchers try to modify the membrane's structure to improve this aspect. Shao and his group [13] created a positively charged membrane. The base of the NF membrane was a polysulfone (PSU) UF membrane, modified by the addition of the polyethyleneimine (PEI) and graphene oxide (GO) active layers. Membranes had a 70 $cm^2$ effective area and various concentrations of GO. Using the membrane with the highest concentration of graphene oxide (40 ppm) resulted in the highest nickel retention—96%. However, the difference between the other membranes was not significant. Hence, the PEI-GO-enhanced membrane surface is a good direction for nickel separation. Another NF membrane modification was proposed by Qi and co-researchers [10]. In this work, a positively charged NF membrane was prepared by using 2-chloro-1-methyliodopyridine as an active agent to graft polyimide polymer onto the membrane surface via covalent bonding with surface carboxylic groups. The results exhibited a high removal efficiency for toxic heavy metal ions. In the case of $NiCl_2$, this value reached 95.8%. Zareei and Mohren Hosseini [14] proposed the use of cobalt ferrite ($CoFe_2O_4$) as a nanomaterial to improve the polyethersulfone membrane (PES) with a MWCO of 58 kDa. Membranes with different concentrations (0.05%, 0.1%, 0.5%, and 1%) of cobalt ferrite were formed using the phase inversion method. The retention of nickel reached 92 and 87% for 0.1 and 0.5% concentrations of cobalt ferrite, respectively. The non-modified membrane rejected the nickel at 79%. Satisfying results obtained by Chunli Liu et al. [11] formed a poly(acrylene ether ketone) (PEAK-COOH) and polyethyleneimine (PEI) membrane by inversion phase and ion complexation. They tested three PEI molecular masses in these experiments and revealed that the 1 kDa mass resulted in the highest retention of nickel—98%.

### 2.2. Reverse Osmosis

Reverse osmosis (RO) has become more attractive for the treatment of industrial wastewater because of its highly efficient and easy operation. There is a lot of research on removing heavy metals by RO. This process can depend on many factors, like pressure, pH, temperature, and others. Using RO membranes is associated with higher operation costs than NF due to the higher transmembrane pressure that has to be applied.

Qin et al. [15] studied the impact of pressure in the range 5–20 bar and pH 3.6–7.0 on the removal of nickel from waste by polyamide RO membranes (with initial conductivity rejections of 96.4, 85.4, and 90.6%). High rejection of nickel ions (93.9, 95.1, 96.7, and 96.8%) was observed for 5, 9, 15, and 20 bars, respectively. The study also revealed that nickel retention increased significantly with increasing pH values. The drop in pH from 3.6 to 7.0 improved the rejection of nickel from 82.1 to 96.4% (TMP = 15 bar). Algureiri et al. [16] also noticed that pH and transmembrane pressure can improve nickel retention. Additionally, they delivered information about the temperature influence on the separation and noticed that at higher temperatures, nickel passes through the membrane structure more readily. A total retention rate of almost 99% was obtained. Ates et al. [17] used a RO polyamide membrane for the removal of heavy metals from aluminum anodic oxidation wastewaters. Tests were performed for 12 h, and nickel was almost completely removed

(99%) after applying 10 and 20 bars. On the other hand, a relevant decrease in permeate flux was observed at 44% for 10 bars and 40% for 20 bars, which is a disadvantage due to the low level of permeate stream in RO separation. To prevent fouling and decrease flux, membranes were chemically cleaned. Unfortunately, between 71 and 90% of the original stream values were recovered. The significant drop in the permeate flux is one of the most problematic issues in this type of separation.

Fouling prevention in RO processes is pre-treated by an ultrafiltration (UF) membrane. This solution was used by Petrinic et al. [18]. The polyvinylidene fluoride UF membrane (transmembrane pressure TMP—2.5 bar) was equipped as a pre-cleaning system in metal-contaminated wastewater treatment. Additionally, besides further increasing the permeate flux, this process removed 15% of nickel. The complex UF-RO process resulted in total (100%) retention of nickel.

A quite innovative idea seems to be the combination of reverse osmosis with chelation. Mohsen-Nia et al. used a disodium salt ethylene diamine tetraacetic acid ($Na_2EDTA$) as a complexing agent and a polyamide membrane for the removal of $Cu^{2+}$ and $Ni^{2+}$ from wastewater. The sewage was pre-cleaned with granular activated carbon and a polypropylene sediment filter cartridge. The addition of $Na_2EDTA$ in a 1:1 ratio with nickel ions contributed to the removal of 99% nickel ions using 3 bar TMP [19]. Similar results were obtained by Rodrigues et al. [20]. They also used EDTA as a complexing agent (1:1) and a polyamide-urea membrane to treat this type of wastewater. They applied 5 bars and removed 99% of the nickel ions. They also noticed that RO filtration with EDTA contributes to increasing permeate flux (about 20 L/h·m² after 8 h of treatment).

In the case of both NF and RO, the obtained permeates are purified water that can be reused in the process [21]. In addition, it is always worth considering reusing the retentate from these purification processes, as it contains large amounts of nickel (and other metals). Such solutions can be reused in the electroplating process or concentrated to recover a specific metal [22].

## 2.3. Complex Ultrafiltration

Complexing ultrafiltration is a quite distinct membrane method used for water purification. Because of complexing compounds, it is possible to successfully separate heavy metal ions ($Cu^{2+}$, $Zn^{2+}$, $Cr^{3+}$, $Ni^{2+}$, $Pb^{2+}$, $Cd^{2+}$, etc.) in ultrafiltration processes, although these membranes are rather applied to much bigger particle separations, e.g., proteins [23]. The complex of a given polymer, e.g., polyethylenoimine and nickel, gives a greater-size structure (Figure 1).

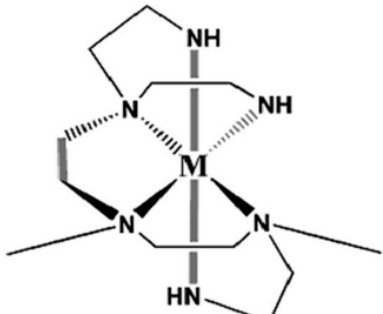

**Figure 1.** Schematic illustration of the chemical coordination of a metal ion and polietyleneimine [24].

To achieve the high efficiency of this process, it is very important to select a suitable complexing agent. It should have good water solubility and chemical stability and be able to form stable complexes with given ions.

Shao et al. [25] used a polyethersulfone membrane (MWCO 30 kDa) and two different complexing agents to purify water from nickel ions at pH = 7 with a polymer:$Ni^{2+}$ ratio of 5:1. The nickel rejections were received with sodium polyacrylate (99.5%) and polyethyleneimine (93.0%) applications. Polyethyleneimine was also successfully used by

Molinari et al. [26] to separate nickel and copper ions from aqueous solutions. They used polyethersulfone membranes with two different MWCO values: 10 and 30 kDa. Separation of copper and nickel ion solutions was carried out under slightly acidic conditions (pH = 6) using polyethyleneimine at a concentration of 150 mg/L (in a ratio of 3:1 in accordance with both metal concentrations). The total nickel rejection (100%) was achieved for a membrane with MWCO 30 kDa. Whereas, a more suitable membrane for copper separation was 10 kDa with a rejection of 95–97% (depending on the transmembrane pressure).

These results were confirmed by another work presented by Molinari et al. [27]. In this study, the copper ions were complexed by polyethyleneimine and separated by six different membranes: polyethersulfone membranes with MWCO of 10, 30, and 40 kDa; polypropylene fluoride membrane (MWCO—40 kDa); polyacrylonitrile membrane (MWCO—40 kDa); and polysulphone-polypropylene (MWCO—40 kDa). The highest retention of copper was achieved at pH = 6 for the polyacrylonitrile membrane (R = 95%). Moreover, they observed that membrane fluxes slowly decreased with an increase in the polymer loading in the solution, which was probably related to the viscosity of the feed.

Kochkodan et al. [28] checked the effect of the different polymer and pH conditions on the separation of copper ions. They used dextrans with different molecular weights (10 and 70 kDa), polyethyleneimine, and sodium lignosulfonates with a copper ion/polymer mass ratio of 1:10 and 1:1. The experiments were carried out using an UMP-20 membrane with 17 kDa MWCO. The pH of 5.1 was indicated to be most effective for polyethyleneimine (97.7% rejection of copper). They proved that the pH solution (in the case of complexation application) was crucial. At pH 2.5, only 35% of the copper ions were chelated by polyethylenimine, whereas after increasing pH to 5.0, the complexation of ions reached 93%.

A copolymer of maleic acid and acrylic acid is another synthetic polymer used as a complexing agent for heavy metal ions in the work by Qiu and Mao [29]. At pH = 6, the capacity of 1 g of this copolymer was able to complex heavy metals from 0.14 to 0.17 g. The separation was carried out using a hollow-fiber polyvinyl butyral membrane with a MWCO of 20 kDa. Satisfactory metal purification of the solution was achieved for each of the test ions ($Cu^{2+}$ 99.8%, $Zn^{2+}$ 98.8%, $Ni^{2+}$ 99.0%, and $Mn^{2+}$ 99.6%). Moreover, the filtrate flux decreased only slightly during separation—only 0.6%.

A slightly different approach involves using environmentally friendly polymers. Lam et al. [30] used chitosan and carboxymethylcellulose, which can be alternatives to the previously mentioned substances. The complexation ultrafiltration process was carried out at pH 5.4 using a polyamide membrane with a MWCO of 3.5 kDa. However, the separation results were not so satisfying in comparison to the previously applied agents. For both natural complexing substances, it was approx. 60%. Moreover, they notice that the presence of chitosan in the solution causes a significant flux decrease. This may be associated with an increase in viscosity or an accumulation of the polymer on the membrane surface. The use of more eco-friendly polymers (of natural origin) for this process is obviously better for the environment; however, it results in worse retention (indirectly—because initially it most likely results in a lower degree of complexation) and a greater decrease in the permeate flux—generally worse process parameters.

The disintegration of complexes during the separation process is a challenge to overcome. This leads usually to release ions to the solution again. The high shear forces created by the pump connected with the membrane module often cause this. To prevent this, Gao et al. [31] used a polysulfone membrane module (MWCO 10 kDa) with a rotating disk to separate nickel ion complexes from wastewater. This element allows for controlling the shear forces by adjusting its revolutions per minute. This technological solution gave satisfying results after complexation with sodium polyacrylate, and researchers received 98% rejection of nickel ions with total stability. Similar results were obtained by Tang and Qiu [32] during zinc ion separation (rejection above 95.3%). They also used sodium polyacrylate as the complexing agent and a polysulfone membrane module (MWCO 30 kDa) with a rotating disk. In this case, the polymer-to-metal ratio was very high—25. The same complexing agent and similar separation conditions were applied in the work of Le and

Qiu [33] to treat wastewater from heavy metals: cadmium, zinc, and lead. In this case, the process conditions were pH = 7, a polysulfone membrane with MWCO of 20 kDa, and a ratio of polyacrylate to metal ions of 10. The concentrations of metals were lower than 0.1 mg/L since the initial concentrations [in mg/L]: Pb(II) 73.2, Cd(II) 163, and Zn(II) 89.1.

### 2.4. Liquid Membranes

Another interesting and effective type of membrane in nickel (metals) recovery are liquid membranes. They can form self-supporting membranes as double emulsions or be immobilized in the pores of another porous membrane (SLM—supported liquid membranes). Thereby, they separate the aqueous solution with a high nickel content (treated wastewater) from the aqueous purification solution. The membrane is usually an organic, water-immiscible solution or ionic liquid [34]. However, their application is quid complicated and, in comparison to the rest of membrane technics less efficient.

Molinari et al. [35] used the SLM system (Figure 2) and indicated the most efficient carrier as 2-hydroxy-5-dodecylbenzaldehyde (2H5DBA) in kerosene for the copper (a metal from the same group as nickel) separation. They compared it to other carriers such as di-(2-ethylhexyl) phosphoric acid (D2EHPA). The experimental data showed that application of 2H5DBA gave a lower copper flux (8.67 mmol/(h·m$^2$) vs. 36.71 mmol/(h·m$^2$), a shorter lifetime (20 h vs. 57 h), and a lower mass transfer coefficient in the film (3 × 10$^{-7}$ m/s vs. 2 × 10$^{-6}$ m/s). Summarizing, the quantity effects are better in the case of 2H5DBA; however, higher selectivity of the separation process was received using D2EHPA.

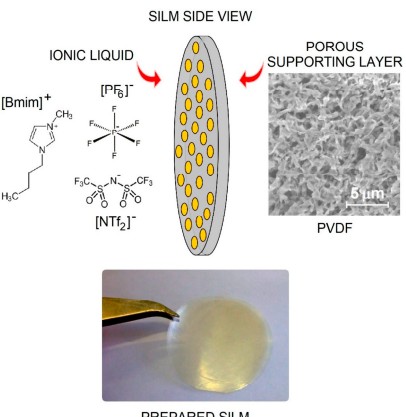

**Figure 2.** The SLM system [36].

D2EHPA in the SLM system was used by Srirachat et al. [37] as well. In this case, they enhanced D2EHPA with TBP (tributyl phosphate) and used a polypropylene hollow fiber microporous membrane. This combination of D2EHPA/TBP in a 1:1 ratio (M:M) made it possible to achieve 85.7% nickel extraction from the wastewater.

The D2EHPA was quite often tested toward nickel recovery, and the researchers tried to improve this diester efficiency or compare it to other possible recovery materials. Yesil et al. [38] compared it to Aliquat 336 (also in the SLM system with a microporous hydrophobic polyvinylidene fluoride (PVDF) membrane). However, in both configurations, the metal removal efficiency was quite low and reached 27.1 ± 1.3% and 46.0 ± 4.3% in the applied conditions, respectively. The same membrane (PVDF) was used by Zante et al. [39], but with a different membrane phase—undiluted hydrophobic ionic liquid tri (hexyl) tetradecylphosphonium chloride [P$_{66614}$] [Cl]. Under these conditions, 86% of the metal (cobalt in this case) can be recovered.

The SLM system is quite popular and is being investigated by scientists for issues with liquid membranes. A different approach is the ELM (emulsion liquid membrane) system, which usually demands surfactant to maintain the double emulsion (Figure 3).

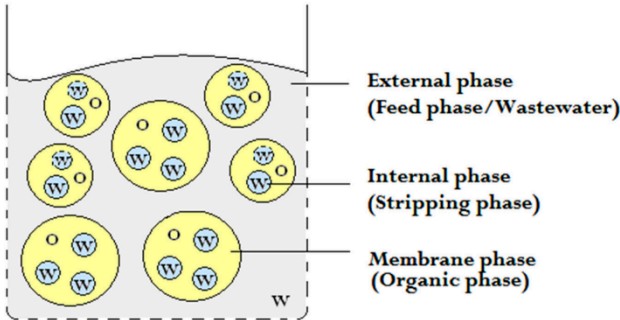

**Figure 3.** The ELM system [40].

This type of separation was investigated by Eyupoglu et al. [41]. The scheme consisted of LIX 63 (8-diethyl-7-hydroxydodecan-6-one oxime) and 2BDA (2-bromodecanoic acid) diluted by kerosene. As it was pointed out, this system demands surfactants to create the membrane phase (Figure 3); the researchers used Span 80 for this purpose. The experiment gave satisfying results. Namely, 98.5% of nickel was removed from the Cr-Ni electroplating bath solution in 6 min. Hachemaoui et al. [42] used ELM that consisted of Cyanex 301 as the extractant. The results of the solvent extraction showed that almost 99% of the cobalt and nickel were extracted with 0.1 M Cyanex 301 at a pH above 2 in just 2 min. As it was presented, the ELM system is even more effective in nickel recovery (very often near 100%), but there is also a difficulty with increasing the scale—mostly used in the laboratory range.

*2.5. Donnan Dialysis*

Donnan dialysis (DD) is another useful technique for separating nickel and other heavy metals from industrial wastewater. It uses ion-exchange membranes (anion-exchange or cation-exchange) that separate two electrolyte solutions, allowing ions of the same valence to exchange between them. The driving force behind the process is the chemical potential gradient on both sides of the membrane. Ions (anions or cations) presented in the feed solution, where their concentration is low, are transported through the membrane with the appropriate charge to the concentrated receiving solution. To ensure the electrical neutrality of the solutions, ions of the same valence in the receiving solution are transported in the opposite direction. The migration of ions between solutions continues until the Donnan equilibrium is reached [43]. Compared to other methods, Donnan dialysis is economically and energetically favorable because it does not require high pressures or electricity [44]. Because DD separation works in a continuous system, the researchers try to develop optimal conditions where the permeability of the metal ions is the highest.

Ersoz, M., and H. Kara [45] studied the transport of $Co^{2+}$ and $Ni^{2+}$ ions through two ICE-450 polysulfonated membranes (homogeneous, unsupported SA3S and heterogeneous, polyester-supported SA3T) as a function of pH gradient. The $H_2SO_4$ solution with a pH range of 1–4 was used as the receiving solution. The values of the diffusion coefficients of the studied polysulfone membranes varied from $4.8 \times 10^{-6}$ to $6.4 \times 10^{-6}$ cm$^2$/s depending on the pH gradient. The results obtained by the authors suggest that the flux of metal ions through the polysulfone ionic-exchange membranes increases when the pH value decreases or when the concentration of $H_2SO_4$ increases. A different study was presented by Hsu et al. [46]. The ionic permeability of $Zn^{2+}$, $Cu^{2+}$, and $Ni^{2+}$ through the Nafion 115 membrane with a sulfonate functional group was evaluated. As a result of the study, it was found that as the concentration of the feed solution increases, the ionic permeability increases. Moreover, the ion permeability was higher in the single solution than in the binary mixture due to the co-ion competition for the sulfonate groups in the membrane structure. Wan, Dongjin, et al. [47] used the GEFC-107 proton-exchange membrane for the removal of $Cu^{2+}$ ions. As the counterion Na$^+$ was selected. In this study, it was confirmed that the transport of $Cu^{2+}$ from donor to receiver phase increased with increasing Na + R/Cu$^2$ + F molarity ratio and $Cu^{2+}$ concentration in the feed solution. The highest $Cu^{2+}$ removal efficiency was 95.31% at a molarity ratio of 20:1. It was also proven that lowering

the pH of the feeding solution or increasing the pH of the receiving solution increases the transport of $Cu^{2+}$ ions.

Another solution proposed by Koseoglu et al. [48] involves the preparation and application of a P2FAn/PVDF composite cation exchange membrane for the removal of $Cr^{3+}$ and $Cu^{2+}$ ions. Different dopant anions such as p-toluenesulfonate (PTS), 1,3 (6 or 7)-naphthalene trisulfonic acid (NSA), o-aminobenzen sulfonic acid (ABS), and sodium dodecyl sulfate (SDS) were used. For all studied composite membranes, the flux of $Cu^{2+}$ ions was higher than that of $Cr^{3+}$ ions. The highest $Cu^{2+}$ ion flux value was 332 mol/cm$^2$ s, while the recovery factor reached a value of about 75% (with SDS addition).

A different approach was implemented by Wang, Jau-Kai, and C-P. Chang [49]. In their work, the effect of the addition of a complexing agent on the selective transport of a ternary Cu-Ni-Co ion system was investigated. Citric acid, malonic acid, and oxalic acid were used as complexing agents, whereas Selemion CMV with a sulfonate ion exchange group was used as the cation exchange membrane. The obtained results indicated that the addition of complexing agents enhances the selectivity of metal ion transport in the ternary system. Malonic acid was found to be the most suitable agent, and the optimal values of dimensionless permeation fluxes (PCu-Na/PNi-Na/PCo-Na) were found to be 0.75/3.75/5.4.

*2.6. Electrodialysis*

The second membrane technic, which utilizes ion-exchange membranes, is electrodialysis (ED). In comparison to Donann dialysis, this process uses an electric potential to separate ions from a feed solution. An electrodialysis apparatus consists of a negatively and positively charged anode and ion-selective membranes between them. This device can have only one pair of membranes or many (usually alternating) cation- and anion-exchange membranes arranged in an alternating manner. Due to the charges on the electrodes, positively charged ions will migrate towards the cathode; however, if they meet an anion-exchange membrane on their way, they will remain in this chamber, which will result in their concentration in this place. It is a relatively economical process, as it only requires voltage generation. Undoubtedly, one of the major advantages of this process is that there is no need to generate a transmembrane pressure difference. Depending on what the main purpose of the separation is (purification, recovery, or both), it is possible to modify the device accordingly.

Benvenuti and his group [50] focused on the selection of appropriate electrodialysis parameters in order to remove nickel ions. In order to protect the anode, they decided to arrange the membranes in the ACAA (A-anion exchange, C-cation exchange) configuration, thanks to which they reduced the chance of nickel ions deposition on the electrode. Ultimately, they managed to achieve 97.43% nickel removal, and the purified water could be reused in the electroplating process.

It is also possible to use more selective membranes, as Min et al. [51]. This research group used five pairs of membranes with a total area of 275 cm$^2$ and determined the most optimal parameters for nickel and copper removal from aqueous solutions. At a voltage of 12 V, the CMX-SB and AMX-SB membranes managed to remove over 99% of metal ions.

The disadvantage of electrodialysis is its low selectivity, since the positively charged ions will migrate together towards the cathode. Complexing compounds are used to enable the separation of the homonymous ions. Babilas and Dydo [52] showed in their work that the use of citric acid allows the separation of iron from copper ions. On the other hand, interesting results from complexing compounds were presented by Chan et al., [53]. They showed that EDTA, depending on the pH of the solution, more willingly forms complexes with various metal ions. In the case of nickel, this pH value is approx. 2, while for cobalt, the pH is approx. 3. It creates the possibility of the selective removal of these ions by choosing the appropriate pH value for the separation. The metal-EDTA complex can be separated by reducing the pH to about 0.5, and the metal can be precipitated as hydroxide upon addition of NaOH. However, Gmar and Chagnes showed that complexing metals

with EDTA can reduce membrane separation efficiency [54], which is quite predictable since the grater complexes can cause an additional fouling layer near the membrane surface.

To test the efficiency and cost-effectiveness of industrial-scale electrodialysis, Ben-evenutia et al. [55] used ACAA electrodialysis (anion exchange membrane—Ionac MC-3475; cation exchange membrane—Ionac MC-3470) to purify the electroplating process wastewater. A month-long experiment showed that it was possible to create a closed water cycle in the treatment process because the purified water had the appropriate quality to be reused for the rinsing process. It has been estimated that the use of electrodialysis will save about 150 m$^3$ of water per year, which will translate into a profit of USD 3800.

As it is presented in this review, almost every known membrane technic can be applied during electroplating wastewater treatment. An application of all of them results in a drop in nickel (and other heavy metal) concentration. Table 1 presents examples of nickel removal efficiency for each membrane technic applied in the treatment of this type of waste.

**Table 1.** Comparison of membrane processes for nickel removal from wastewater.

| | | NF | | | RO | | | Complex UF |
|---|---|---|---|---|---|---|---|---|
| Membrane Type | (1) | NF90, NF270, PP-NF | (1) | | Polyamide RO membrane | (1) | | Polyehersulfone UF membrane |
| | (2) | PSU-UF membrane modified by PEI-GO | (2) | | Polyamide RO membrane | (2) | | Polyehersulfone UF membrane |
| | (3) | NF membrane modified by polyimide | (3) | | Polyvinylidene fluoride UF-NF membrane system | (3) | | Polyvinyl butyral UF membrane |
| | (4) | PES-NF membrane modified by cobalt ferrite (0.1%, 0.5%,) | (4) | | Polyamide membrane with chelating agents | (4) | | Polyamide UF membrane |
| | (5) | PEAK-PEI NF membrane | (5) | | Polyamide-urea membrane with chelating agents | | | |
| Retention of metal | (1) | 98, 70, 75% | (1) | | From 82.1 to 96.8% (depends on pH and TMP) | (1) | | 99.5% with polyacrylate and 93.0% as polyethylenamine as a complex agent |
| | (2) | 96% | (2) | | 99% | (2) | | 100% (with membrane with 30 kDa and polyethylenamine as a complex agent |
| | (3) | 96% | (3) | | 100% | (3) | | 99.0% (Copolymer of maleic acid and acrylic acid) |
| | (4) | 92 and 87% | (4) | | 99% | (4) | | 60% (chitosan and carboxymethylcellulose) |
| | (5) | 98% | (5) | | 99% | | | |
| Reference | (1) | [12] | (1) | | [15] | (1) | | [25] |
| | (2) | [13] | (2) | | [17] | (2) | | [26] |
| | (3) | [10] | (3) | | [18] | (3) | | [29] |
| | (4) | [14] | (4) | | [19] | (4) | | [30] |
| | (5) | [11] | (5) | | [20] | | | |
| | | LM | | | DD | | | ED |
| Membrane Type | (1) | D2EHPA/TBP in a polypropylene hollow fiber microporous membrane | | | | | | |
| | (2) | SLM system with microporous hydrophobic polyvinylidene fluoride (PVDF) membrane with Aliquat 336 and D2EHPA | (1) | | GEFC-107 proton-exchange membrane | (1) | | membranes in the ACAA configuration |
| | (3) | ELM system with LIX 63 (8-diethyl-7-hydroxydodecan-6-one oxime) and 2BDA (2-bromodecanoic acid) diluted by kerosene | (2) | | P2FAn/PVDF composite cation exchange membrane | (2) | | CMX-SB and AMX-SB membranes |
| | (4) | ELM with Cyanex 301 | | | | | | |
| Retention/Removal/Recovery of nickel | (1) | 85.7% | | | | | | |
| | (2) | 46.0 and 27.1%, respectively | (1) | | 95.31% * | (1) | | 97.43% |
| | (3) | 98.5% | (2) | | 75.0% * | (2) | | 99.0% |
| | (4) | 99.0% | | | | | | |
| References | (1) | [37] | | | | | | |
| | (2) | [38] | (1) | | [47] | (1) | | [50] |
| | (3) | [41] | (2) | | [48] | (2) | | [51] |
| | (4) | [42] | | | | | | |

* removal of Cu$^{2+}$.

All of the mentioned technics have some disadvantages and advantages (Table 2). However, there are two most appropriate technics, due to their higher effectiveness than others.

**Table 2.** Comparison of membrane technics used in electroplating wastewater treatment.

| | Advantages | Disadvantages |
|---|---|---|
| Nanofiltration | High level of metal retention | High value of TMP (high energy consumption)<br>Scaling<br>Necessity of polymeric membranes application |
| Reverse osmosis | Very high level of metal retention | Very high value of TMP (high energy consumption)<br>Low value of permeate stream<br>Scaling<br>Necessity of polymeric membranes application |
| Complex Ultrafiltration | High value of permeate stream<br>High level of metal retention<br>Possibility of ceramic membranes application (longer lifespan)x<br>Lower (than NF and RO) TMP (lower energy consumption) | Fouling<br>Necessity of complex agent application |
| Liquid membranes | High selectivity | Lower than other efficiency<br>Necessity of organic solutions application<br>Difficulties of process performing |
| Donan Dialysis | Low energy consumption | Lower than other efficiency |
| Eletrodialysis | High efficiency (high stream of treated wastewater and high retention of metal)<br>No pressure technics | Complex apparatus<br>Expensive (ion-exchange) membranes |

The first is based on the possibility of forming complexes by nickel complexing ultrafiltration, which combines relatively high permeate flux values with high selectivity. The second, using the fact that nickel exists in the form of cations, is electrodialysis. ED, similarly to CUF, leads to large streams of treated wastewater and a high nickel separation coefficient.

## 3. Future Perspectives

As it was presented above, many different membrane technics can be successfully applied to the wastewater derived from electroplating process treatment. One is more efficient than the other. Based on the presented review, one can draw the conclusion that electrodialysis and complexation ultrafiltration are the most adequate, so they are deeply investigated by scientists. The second aspect that should be improved and developed in detail in this issue is membrane material. The structure of it should be enhanced toward higher nickel or nickel complex retention, higher permeate flux, or lower fouling ability.

A quite new approach to membrane preparation is the application of nanomaterials for this purpose. Nanocomposite membranes are the most promising solution for metal recovery from wastewater since there is a possibility of a connection between the membrane's ease of operation and the high absorptive selectivity of nanoparticles.

### 3.1. Complexation Ultrafiltration

Pressure-driven processes such as micro- and ultrafiltration have too large pore sizes to retain the nickel ions. However, these types of membrane separations are favorable in industrial wastewater treatment since they generate the greatest permeate fluxes. Complexation ultrafiltration is nowadays a widely developed issue. This technics connects the high flux of ultrafiltration with the retention of nanofiltration.

Due to the huge emphasis on green technologies in every branch of industry, this trend has recently dominated wastewater treatment as well. The complex agents should be eco-friendly and biodegradable. Hence, the researchers try to find agents that are efficient and not harmful to the environment at the same time. A sodium salt ethylene diamine tetraacetic acid (Na2EDTA) [19], polyacrylic acid sodium (PAAS) [32] or polyethylene imine

(PEI) [56] is commonly used as a chelating agent. However, they are not recognized as eco-friendly. Now biopolymers and nature-derived agents are tested toward nickel (and other metals) complexation, such as chitosan, alginic acid, sodium alginate, chitin, cellulose, or carboxycelullose [57].

The newest agent developed as a green complexation component was presented by Zebat [58]. The objective of his work was to study the removal of nickel using a polyoxometalate ($\alpha$2P2W17O61) [10] as a ligand. The results have shown that if the ligand/metal ratio of the formed complex was one, the formed complex was very stable in an aqueous medium (with a high value of the stability constant). This direction should still be developed and investigated—finding high-efficiency, eco-friendly complexity agents creates stable complexes.

### 3.2. Nanocomposite Membranes

A completely different approach is the application of new-generation nanocomposite membranes in nickel-contaminated wastewater treatment. During the formation of these types of membranes, nanomaterials are introduced to the thin layer of composite membrane (Figure 4).

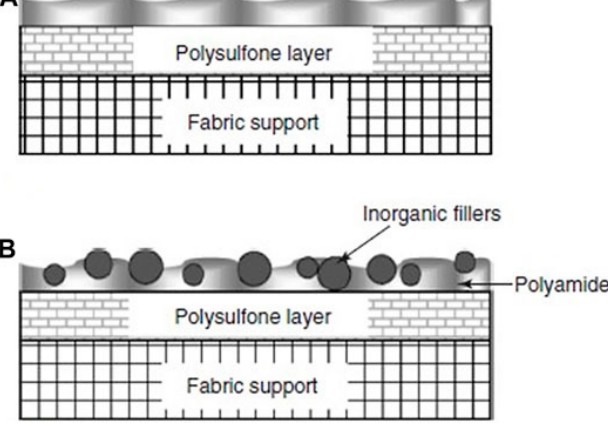

**Figure 4.** Concept of nanocomposite membranes. (**A**) without the nanomaterials; (**B**) with the introduced nanoparticles [59].

This modified surface can adsorb heavy metals; hence, the separation effect of, e.g., a nanofiltration membrane is enhanced by adsorption [59,60]. The most popular nanoparticles used in membrane surface modification are $Fe_2O_4$, graphene oxide (GO), polyaniline, $Al_2O_3$, and titanate nanotubes (TNT). The modified skin layer is usually prepared separately from the support phase by inversion or interfacial polymerization method [61]. Applying this approach, Moradihamedani and his research group prepared and tested a $Fe_3O_4$-talc nanocomposite incorporated into a polysulfone membrane that resulted in 96.2% nickel rejection from a 50 mg/L $Ni^{2+}$ aqueous solution. Activated bentonite clay nanoparticles embedded on polyetherimide membranes were able to reject 76.2% and 64.8% $Ni^{2+}$ from 250 and 1000 ppm feed solutions, respectively [62].

### 3.3. Electrodialysis

The ED described above is one of the most efficient membrane processes in the case of $Ni^{2+}$ and other metal recovery, since this is the most appropriate technic for the ion separation. This separation method is already used in the wastewater treatment industry; however, it should be improved all the time to increase the efficiency of the process. This can be associated with the membrane itself or with a whole ED system modification. The importance of wastewater pretreatment was highlighted to prevent the fouling of the ED membrane surface due to organic or pollutant deposition [63]. This is an important

consideration to make before working with real industrial effluents to enhance the lifetime of the system and reduce operating costs [64].

A quite new approach is the selectrodialysis (SED) process, which is usually employed in metal-contaminated wastewater treatment to selectively separate metals in different ED compartment streams. In this section, the equilibrium of the chemical species plays a critical role since it allows the effective separation of certain metals. This process overcomes one of the main limitations of ED, which is its low selectivity for the recovery of specific metals. The isolation is achieved by different pH, ionic charge, and chelating/complexation agent applications [64]. Due to the high selectivity of the process, a highly selective membrane should be produced. This is a very often developed issue—modification of anion-cation-exchange membranes to improve metal separation properties.

## 4. Conclusions

Nanofiltration is a separation technique that uses membranes that range from RO membranes that potentially retain nickel to UF membranes, which have too large pores for this purpose. It means that the retention of nickel depends on the type of NF membrane, especially its MWCO. Nickel retention can be increased by improving the adsorption properties of the skin layer of the NF membrane (these are usually composite membranes). NF is the most appropriate technic from the nonenhanced pressure-driven technic, since it combines a sufficient level of flux and retention of metal, and nowadays it is the most reasonable to apply it as a final step in electroplating wastewater treatment.

The mentioned reverse osmosis retains nickel from the electroplating wastewater, but its use is associated with relatively low values of the permeated flux and its drop during filtration. Using the complexing RO makes it possible to use a lower value of transmembrane pressure (due to the lower osmotic pressure of the wastewater). The application of this technic is reasonable if the final stream of treated wastewater is of very high quality. Whereas, complexing ultrafiltration is one of the most promising solutions for the treatment of water from the electroplating process. Mainly, in terms of reducing the content of metals such as nickel, chromium, and zinc. It combines the high permeated fluxes of classical ultrafiltration with the high selectivity of nanofiltration. The current challenge is to find an eco-friendly complexing agent, and many researchers try to find a naturally derived complexing agent, e.g., chitosan or carboxymethylcellulose, that will be as efficient as these typical chemical agents, e.g., acrylic acid.

Donnan dialysis is a process typically dedicated to the separation of ions and therefore ideally suited to the separation of nickel. Due to the fact that it is a process in which there is a concentration difference as a driving force at the beginning, it is a low-energy process and thus rather inefficient. The use of ion exchange membranes with a current source (electrodialysis (ED)) significantly improves and intensifies this process, and as a result, ED is the second most popular process in electroplating wastewater treatment. Currently, a new direction in the treatment of this type of wastewater is the development of selective electrodialysis (SED), which may additionally lead to the separation of a specific metal ion, e.g., nickel. This is a more advanced approach to the treatment of electroplating wastewater. It leads to a lower concentration of harmful substances in the output stream and the recovery and reuse of the metal used in the main industry processes. This can reduce the capital cost of production.

**Author Contributions:** Conceptualization, M.L.; investigation, M.L., O.G., K.H., K.K., H.K., Ł.M. and E.S.; writing—original draft, M.L., O.G., K.H., K.K., H.K., Ł.M. and E.S.; preparation, M.L.; writing—review and editing, M.L.; visualization, M.L.; supervision, M.L. All authors have read and agreed to the published version of the manuscript.

**Funding:** This research did not receive any specific grants from funding agencies in the public, commercial, or not-for-profit sectors.

**Data Availability Statement:** Data sharing not applicable. No new data were created or analyzed in this study. Data sharing is not applicable to this article.

**Acknowledgments:** This article was created as part of the authors statutory research activity at the Department of Micro, Nano, and Bioprocess Engineering, Faculty of Chemistry, Wroclaw University of Science and Technology (Poland).

**Conflicts of Interest:** The authors declare no conflict of interest.

## Abbreviations

| | |
|---|---|
| UF | ultrafiltration |
| NF | nanofiltration |
| RO | reverse osmosis |
| ED | electrodialysis |
| CUF | complexing ultrafiltration |
| SED | selective electrodialysis |
| MWCO | molecular weight cut-off |
| PSU | polysulfone |
| GO | graphene oxide |
| PEI | polyethyleneimine |
| TMP | transmembrane pressure |
| EDTA | ethylene diamine tetraacetic liquid |
| R | retention coefficient |
| SLM | supported liquid membranes |
| ELM | emulsion liquid membranes |
| TNT | titanate nanotubes |

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
