# Peer review of "Membrane Separation in the Nickel-Contaminated Wastewater Treatment"

_waste, doi:10.3390/waste1020029_

Round 1

Reviewer 1 Report

1- The authors should be provide an abbreviation list.

2- Advantage and disadvantages of the membrane separation should be provide in the review article.

3- The feasibility of reuse of the treated nickel should be provide in the manuscript.

Author Response

Dear Reviewer,

I would like to thank you for your opinion. Based on yours suggestions there were introduced the modifications to the text as much as possible.

  • The authors should be provide an abbreviation list.

The abbreviation list was added.

  • Advantage and disadvantages of the membrane separation should be provide in the review article.

This literature review refers to all membrane techniques from pressure-driven to those based on electric potential difference. Therefore, it is difficult to make a general list of advantages and disadvantages of membrane techniques in general. We deal with different in case of NF and different to e.g. liquid membranes. Some of them were mentioned in appropriate chapters (described given membrane technic).

  • The feasibility of reuse of the treated nickel should be provide in the manuscript.

This aspect of nickel reuse in the electroplating process was added to the manuscript.

Reviewer 2 Report

Nice work but its needs minor revision.

1.      Rewrite the keywords.

2.      Novelty of the review should be established.

3.      Reported figures are not precise; check them carefully.

4.      The authors should compare the efficiency using supported liquid membranes, emulsion liquid membranes, donnan dialysis, and electro dialysis systems.

5.      The authors should include the comparison table of nickel removal efficiency using the different systems.

6.      Improve the conclusion part.

7.      Check reference 12 carefully.

8.      English language should be improved.

9.      Typo errors in the manuscript. Check it very carefully. 

Author Response

Dear Reviewer,

I would like to thank you for your opinion. Based on yours suggestions there were introduced the modifications to the text as much as possible.

Nice work but its needs minor revision.

  1. Rewrite the keywords

The Keywords were rewritten.

  1. Novelty of the review should be established.

This aspect was introduced to manuscript.

  1. Reported figures are not precise; check them carefully.

We are delighted to indicate the mistakes. The figure 2 was corrected (SILM corrected to SLM). Rest of them seems to be correct, however if there is more incorrectness, we would be glad if they can be indicated by Reviewer.

  1. The authors should compare the efficiency using supported liquid membranes, emulsion liquid membranes, donnan dialysis, and electro dialysis systems.
  2. The authors should include the comparison table of nickel removal efficiency using the different systems.

The comparison was included to the manuscript.

  1. Improve the conclusion part.

The conclusion part was improved as much as possible.

  1. Check reference 12 carefully.

The reference was checked and corrected.

  1. English language should be improved.

The English was improved as much as possible

  1. Typo errors in the manuscript. Check it very carefully. 

The manuscript was read carefully and some errors were found and corrected.

Reviewer 3 Report

The manuscript reviews the application of Membrane separation technology for the treatment of nickel-contaminated wastewater treatment. In the current form of the manuscript, it seems like the authors have prepared a report on the treatment of nickel-contaminated wastewater using Membrane separation. The major weakness of the manuscript is the limited data collection, lack of scientific discussion with the similar and dissimilar findings of the various membrane separation technologies . Therefore, I am not recommending to accept the manuscript in its current form. Herewith I have added some suggestions to improve the manuscript.

Comments:

1. Please add a methodology on how the data was collected from the literature and how many studies were collected to build this current review article.

2. Author may add the various nickel treatment technologies and their limitation in the removal of Ni(II) from nickel-contaminated wastewater. How the membrane separation technology will address the limitations.

3. Please provide data on the experimental conditions of the various membrane separation technologies and their efficiency on Ni(II) separation.

4. Finally, the author may propose the best appropriate membrane separation technology for Ni(II) separation of nickel contaminated wastewater by addressing the advantages and limitations of various membrane separation technologies. 

Author Response

Response to Reviewer 3:

  1. Please add a methodology on how the data was collected from the literature and how many studies were collected to build this current review article.
  2. Author may add the various nickel treatment technologies and their limitation in the removal of Ni(II) from nickel-contaminated wastewater. How the membrane separation technology will address the limitations.
  3. Please provide data on the experimental conditions of the various membrane separation technologies and their efficiency on Ni(II) separation.
  4. Finally, the author may propose the best appropriate membrane separation technology for Ni(II) separation of nickel contaminated wastewater by addressing the advantages and limitations of various membrane separation technologies. 

We would like to thank reviewer for suggestions given. We tried to improve the manuscript as much as possible, however some of them were quite difficulty to introduce. Mainly, including process parameters would expand the work too much and could make it illegible. The most important parameter, which is the efficiency or effectiveness of the process, has been given. The reader interested in details can check the exact parameters in specific publications, which are indicated at the end of the article.

The number of used studies are given in the last chapter, where the literature positions are numbered (there are 64 positions). The literature items were searched in the google scholar database using the keywords "nickel membrane separation" and "membrane techniques in electroplating wastewater treatment".

The most appropriate technics of nickel-contained wastewater were indicated in the last section and the section about the limitations in case of most of them as well.

Round 2

Reviewer 1 Report

The authors have addressed my concerns appropriately, and I recommend the manuscript be now accepted for publication in its current form.

Reviewer 2 Report

The authors have responded to the reviewer's comments and can be accepted. 

Reviewer 3 Report

Satisfied with responses taken by authors. Therefore, I am recommending to accept the manuscript in its present form.